# Processing SPARQL Property Path Queries Online with Web Preemption

Julien Aimonier-Davat, Hala Skaf-Molli, and Pascal Molli

LS2N – University of Nantes, France
{Julien.Aimonier-Davat,hala.skaf,pascal.molli}@univ-nantes.fr

**Abstract.** SPARQL property path queries provide a succinct way to write complex navigational queries over RDF knowledge graphs. However, the evaluation of these queries over online knowledge graphs such as DBPedia or Wikidata is often interrupted by quotas, returning no results or partial results. To ensure complete results, property path queries are evaluated client-side. Smart clients decompose property path queries into subqueries for which complete results are ensured. The granularity of the decomposition depends on the expressivity of the server. Whatever the decomposition, it could generate a high number of subqueries, a large data transfer, and finally delivers poor performance. In this paper, we extend a preemptable SPARQL server with a partial transitive closure operator (PTC) based on a depth limited search algorithm. We show that a smart client using the PTC operator is able to process SPARQL property path online and deliver complete results. Experimental results demonstrate that our approach outperforms existing smart client solutions in terms of HTTP calls, data transfer and query execution time.

## 1 Introduction

**Context and motivation:** Property paths were introduced in SPARQL 1.1 [14] to add extensive navigational capabilities to the SPARQL query language. They allow to write sophisticated navigational queries over Knowledge Graphs (KGs). SPARQL queries with property paths are widely used. For instance, they represent a total of 38% of the entire log of wikidata [5]. However, executing these complex queries against online public SPARQL services is challenging, mainly due to quotas enforcement that prevent queries to deliver complete results [11, 9, 10]. In this paper, we focus on *how to execute SPARQL property path queries online and get complete results?*

**Related Works:** The problem of executing property path queries online and get complete results have been already studied in the context of Triple Pattern Fragment (TPF) [17, 8] and Web Preemption [1]. Current approaches decompose property path queries into many triple pattern or BGP queries that are guaranteed to terminate. However, such an approach generates a large number of queries which significantly degrades performance.

**Approach and Contributions:** In this paper, we extend a preemptable SPARQL server with a preemtable Partial Transitive Closure (PTC) operator based on

```
PREFIX wd: <http://www.wikidata.org/entity/>
PREFIX wdt: <http://www.wikidata.org/prop/direct/>
SELECT ?creativeWork ?fictionalWork  WHERE {
  ?creativeWork  wdt:P144 ?fictionalWork  .
  ?creativeWork  wdt:P31/wdt:P279* wd:Q17537576  .
  ?fictionalWork  wdt:P136 wd:Q8253}
```

(a) $Q1$: Creative works and the list of fiction works that inspired them on Wikidata

```
@prefix owl: <http://www.w3.org/2002/07/owl#>
@prefix foaf: <http://xmlns.com/foaf/0.1/>
select ?x ?o where {
  ?x foaf:name ?n .
  ?x owl:sameAs* ?o .
}
```

(b) $Q2$: List of similar entities on DB-Pedia

Fig. 1: Property path queries on online knowledge graphs

a depth limited search algorithm. We show that a smart client using the PTC operator is able to compute SPARQL 1.1 property path queries online and get complete results without decomposing queries. In this paper: (i) We show how to build a PTC preemptable operator. (ii) We show how to build a smart-client that computes transitive closures from partial transitive closures. (iii) We compare the performances of our PTC approach with existing smart client approaches and SPARQL 1.1 servers. Experimental results demonstrate that our approach outperforms smart client approaches in terms of query execution time, number of HTTP calls and data transfer.

This paper is organized as follows. Section 2 reviews related works. Section 3 introduces web preemption and property paths. Sections 4 presents the PTC approches and algorithms. Section 5 presents our experimental results. Finally, the conclusion is outlined in Section 6.

## 2   Related Works

Property paths closely correspond to regular expressions and are crucial to perform non-trivial navigation in knowledge graphs. Regular expressions involve operators such as ' * ' (transitive closure operator, zero or more occurrences-kleene star), ' | ' (OR operator), ' / ' (sequence operator), ' $\wedge$ ' (inverse operator), ' ! ' (NOT operator) that allow to describe complex paths of arbitrary length. For instance, the query `SELECT ?x ?y WHERE { ?x foaf:knows* ?y }` requires to compute the transitive closure of the relation `foaf:knows` over all pairs $(x, y)$ present in a knowledge graph.

***SPARQL Endpoints.*** Many techniques [12, 4] allow to evaluate property paths. There is currently two main approaches: graph traversal based approaches and recursive queries. Whatever the approach we consider, path queries with transitive closures are challenging to evaluate for online Knowledge Graphs such as DBPedia or Wikidata. To ensure a fair usage policy of resources, public SPARQL endpoints enforce quotas [6] in time and resources for executing queries. As queries are stopped by quotas, many queries return no results or partial results. For instance, the query $Q1$ in Figure 1 returns no result on Wikidata because it has been stopped after running more than 60s. The query $Q2$[1] on DBPedia

---

[1] $Q1$ and $Q2$ are respectively executed at the public SPARQL endpoints of Wikidata and DBPedia, at August 5 2020.

returns partial results because it has been killed after delivering the first 10000 results.

***Decomposition and restricted interfaces approaches.*** The problem of executing property path queries online and get complete results have been already studied with SPARQL restricted interfaces represented by Triple Pattern Fragment (TPF) [17, 8] and Web Preemption [1]. As a TPF server [8, 17] ensures the termination of any triple pattern query, a TPF smart client [15] decomposes the evaluation of a property path into multiple triple pattern queries that are sent to the server. This requires to compute several joins on the client, especially to compute transitive closure expressions. This generates many calls and a huge data transfer, resulting in poor performances as pointed out in [1]. As preemptable SPARQL server [10] ensures the termination of any BGP query, the SaGe smart client [1] decomposes the query into BGP queries that are sent to the server. As BGP are supported, performances are better than TPF. However, the preemptable server does not support transitive closures. Consequently, to process the query $Q1$ of Figure 1a the smart client breaks the Basic Graph Pattern (BGP) of $Q1$ into 3 triple patterns. The triple patterns are then processed using nested loop joins where joins are performed on the client. The transitive closure is processed using a simple Breadth First Search (BFS) algorithm implemented on client-side. This process remains clearly data-transfer intensive and generates a very high number of calls to the server. As each call has to pay for the network latency, the execution time of the query is dominated by the network costs.

## 3   Web Preemption and Property Paths

*Web preemption* [10] is the capacity of a web server to suspend a running SPARQL query after a fixed quantum of time and resume the next waiting query. When suspending a query $Q$, a preemptable server saves the internal state of all operators of $Q$ in a saved plan $Q_s$ and sends $Q_s$ to the client. The client can continue the execution of $Q$ by sending $Q_s$ back to the server. When reading $Q_s$, the server restarts the query $Q$ from where it has been stopped. As a preemptable server can restart queries from where they have been stopped and makes a progress at each quantum, it eventually delivers complete results after a bounded number of quanta.

However, web preemption comes with overheads. The time taken by the suspend and resume operations represents the overhead in time of a preemptable server. The size of $Q_s$ represents the overhead in space of a preemptable server and may be transferred over the network each time a query is suspended by the server. To be tractable, a preemptable server has to minimize these overheads.

As shown in [10], suspending a simple triple pattern query is in constant time, i.e., just store the last triple scanned in $Q_s$. Assuming that a dataset $D$ is indexed with traditional B-Trees on SPO, POS and OSP, resuming a triple pattern

query given the last triple scanned is in $O(log(|D|))$ where $|D|$ is the size of the dataset $D$. Many operators such as join, union, projection, bind and most filters can be saved and resumed in constant time as they just need to manage *one-mapping-at-a-time*. These operators are processed by the preemptable SPARQL server.

However, some operators need to materialize intermediate results and cannot be saved in contant time. For example, the "ORDER BY" operator needs to materialize the results before sorting them. Such operators are classified as *full-mappings* and are processed by the smart client. For example, to process an "ORDER BY", all results are first transferred to the smart client that finally sort them. If delegating some operators to the client-side allows effectively to process any SPARQL queries, it has a cost in term of data transfer, number of calls to the server to terminate the query, and execution time. Unfortunately, to compute property path expressions with transitive closures we need a server-side operator that belongs to the *full-mappings* operators. DFS Graph-traversal based approaches require to store at least the current path in the graph that can be in the worst case, of the size of the graph. Recursive-queries approaches require to store a temporary relation that is incrementally saturated, and that also cannot be saved in constant time. Currently, a BGP containing a path expression with a closure is fully processed by the smart-client following the decomposition approach described in the related works (cf section 2).

The only way to reduce the number of calls is to extend a preemptable server with a transitive closure operator such that BGP containing path patterns can be processed on server-side. However, algorithms that implement transitive closure such as DFS and BFS are not preemptable, i.e., cannot be suspended and resumed in constant time.

**Problem Statement:** Define an $\alpha$ operator able to compute the transitive closure such that the complexity in time and space of suspending and resuming $\alpha$ is in constant time.

## 4 The Partial Transitive Closure Approach

To compute SPARQL 1.1 property paths online and deliver complete results, our approach relies on two key ideas:

− First, thanks to the ability of the web preemption to save and load iterators, it is possible to implement a Partial Transitive Closure (PTC) operator that can be saved and resumed in $O(k)$. A PTC operator computes the transitive closure of a relation but cuts the exploration of the graph at a depth $k$. Nodes that are visited at depth $k$ are called frontier nodes. However, such an operator is not able to compute property path expressions as defined in SPARQL 1.1, i.e., transitive closures may be incomplete and return duplicates.

− Second, by sending frontier nodes to the smart client, it is possible to restart the evaluation of a property path query from the frontier nodes. Consequently,

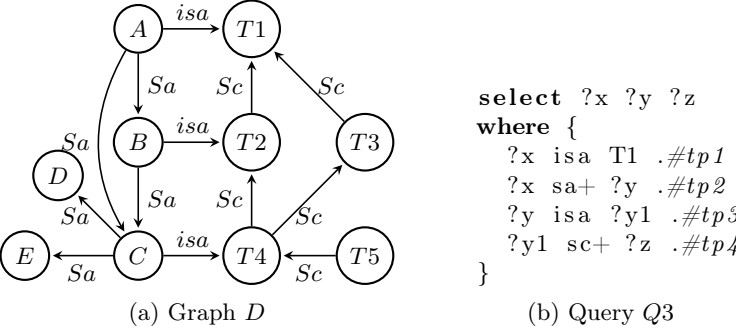

(a) Graph $D$           (b) Query $Q3$

Fig. 2: Graph $D$ and query $Q3$

a smart client using the PTC operator can fully compute SPARQL 1.1 property paths. Such a strategy to compute property paths outperforms smart client approaches as queries are evaluated on the server without any joins on client-side.

In this paper, we focus on the evaluation of transitive path expressions *without* nested stars. For example, property paths like (ab*)* are not considered and will be the subject of future work. We assume that non-transitive expressions such as alternatives or sequences are decomposed and evaluated following [14].

### 4.1 The PTC operator

A *Partial Transitive Closure* $PTC(v, p, k)$ is defined for a starting node $v$, a non-transitive path expression $p$ and a depth $k$. $PTC(v, p, k)$ returns all pairs $(u, d)$ with $u$ a node, such that it exists a path from $v$ to $u$ that conforms to the expression $(p)^d$ where $d \leq k$ and $d$ is minimal.

The *frontier nodes* for a $PTC(v, p, k)$ are the nodes reached at depth $k$, i.e., $\{u \mid (u, d) \in PTC(v, p, k) \wedge d = k\}$.

To illustrate, consider the $PTC(A, Sa, 2)$ that returns all nodes reachable from $A$ through a path that conforms to the expression $(Sa)^d$ with $d \leq 2$. On the graph $D$ of Figure 2a, $PTC(A, Sa, 2) = \{ (B, 1), (C, 1), (D, 2), (E, 2) \}$ where both $D$ and $E$ are frontier nodes. If $PTC(n, p, k)$ returns no frontier nodes then the transitive closure is complete, i.e., $k$ was large enough to capture the transitive closure for parameters $v$ and $p$. Otherwise, frontier nodes are used by the smart client to continue the evaluation of the transitive closure until no new frontier nodes are discovered.

In our context, the depth limit $k$ (*maxDepth*) is fixed by the preemptable SPARQL endpoint administrator and can be seen as a global variable of the preemptable server.

To implement a PTC operator, we rely on a depth-limited search (DLS) algorithm [13] [2]. DLS is fundamentally a depth-first search where the search space is

---

[2] Iterative Deepening Depth-First Search (IDDFS) can also be used, but IDDFS re-traverse same nodes many times.

---

**Algorithm 1:** ALP auxiliary function

---

**1** **Let** $eval(v, p)$ be the function that returns all terms reachable from the RDF term $v$, by going through a path that matches the non-transitive path expression $p$.

**2** **Let** MaxDepth be the depth limit

**3** **Function** *ALP(v:term, p:path)*:

**4**  $\quad R \leftarrow \emptyset$ // set of terms

**5**  $\quad V \leftarrow \emptyset$ // set of pairs (Term, Integer)

**6**  $\quad ALP(v, p, R, V)$

**7**  $\quad$ **return** $R$

**8** **Function** *ALP(v, p, R, V)*:

**9**  $\quad S \leftarrow [(v, 0)]$ // stack of terms

**10** $\quad$ **while** $S \neq \emptyset$ **do**

**11** $\quad\quad (u, d) \leftarrow S.pop()$

**12** $\quad\quad R.add(u)$

**13** $\quad\quad V.add((u, d))$

**14** $\quad\quad$ **if** $d \geq MaxDepth$ **then continue**

**15** $\quad\quad X \leftarrow eval(u, p)$

**16** $\quad\quad$ **forall** $x \in X$ **do**

**17** $\quad\quad\quad$ **if** $\nexists (x, d') \in V,\ d' \leq d$ **then**

**18** $\quad\quad\quad\quad S.add((x, d + 1))$

---

limited to a maximum depth. Algorithm 1 redefines the ALP auxiliary function of the SPARQL 1.1 specification to follow our definition of $PTC$.

To avoid counting beyond a Yottabyte [2], each node is annotated with the depth at which it has been reached (Line 13). A node is revisited only if it is reached with a shortest path (Line 17). Compared to an existential semantics [2] where nodes can be visited only once, the time complexity is degraded because nodes can be revisited at most $k$ times. However, using an existential semantics does not allow to ensure the $PTC$ semantics, as pointed out in [16]. To illustrate, consider $PTC(A, Sa, 2)$ evaluated previously. Under an existential semantics, starting at node $A$, node $B$ is first visited at depth 1, then $C$ at depth 2 and both $B$ and $C$ are marked as visited. As $C$ cannot be revisited at depth 1, $PTC(A, Sa, 2)$ returns pairs $\{(B, 1), (C, 2)\}$. In spite of there is a path from $A$ to $D$ and $E$ that match $(Sa)^2$, nodes $D$ and $E$ are not returned. Moreover, $C$ appears as a frontier node and will be explored by the smart client whereas it is not a frontier node.

### 4.2 pPTC: a preemptable PTC iterator

The most important element of an iterative DLS is the stack of nodes to explore. To build a preemptable iterator based on the DLS, its stack must be saved and resumed in constant time. To achieve this goal, we do not pushed nodes on the stack, but iterators that are used to expand nodes and explore the graph.

Algorithm 2 presents our preemptable Partial Transitive Closure iterator, called $pPTC$. To illustrate how a property path query is evaluated using $pPTC$, suppose the server is processing query $Q3$ with the physical query plan of Figure 3. When the third index loop join iterator is first activated, it pulls the bag of mappings $\mu = \{\ ?x \mapsto A,\ ?y \mapsto C,\ ?y1 \mapsto T4\ \}$ from its left child. Then, it applies $\mu$ to $tp4$ to generate the bounded pattern $b = $ `T4 Sc+ ?z`, creates a $pPTC$ iterator to evaluate $b$ and calls the $GetNext()$ operation of the $pPTC$ iterator, i.e., its right child.

**Algorithm 2:** A *Preemptable PTC Iterator*, evaluating a kleene star expression without nested stars

**Require:**
$p$: path expression without stars
$v$: RDF term
$\mu$: set of mappings
$tp_{id}$: path pattern identifier
**Data:**
$MaxDepth$: depth limit
$S$: empty stack of preemptable iterators
$V$: set of pairs (RDF term, Integer)
$CT$: empty set of control tuples

**1 Function** *Open()*:
**2**  $\quad$ iter $\leftarrow$ createIter($v$, $p$, ?$o$)
**3**  $\quad$ S.push(iter.save())
**4**  $\quad$ $\mu_c \leftarrow$ nil
**5**  $\quad$ iter $\leftarrow$ nil

**6 Function** *Save()*:
**7**  $\quad$ **if** *iter* $\neq$ *nil* **then**
**8**  $\quad\quad$ S.push(iter.save())
**9**  $\quad$ **return** S, path, $tp_{id}$, $\mu_c$

**10 Function** *Load(S', path', tp$_{id}$', $\mu_c'$)*:
**11**  $\quad$ S $\leftarrow$ S'
**12**  $\quad$ path $\leftarrow$ path'
**13**  $\quad$ $tp_{id} \leftarrow tp_{id}$'
**14**  $\quad$ $\mu_c \leftarrow \mu_c'$

**15 Function** *GetNext()*:
**16**  $\quad$ **while** $S \neq \emptyset$ **do**
**17**  $\quad\quad$ **if** $\mu_c = nil$ **then**
**18**  $\quad\quad\quad$ **while** $\mu_c = nil$ **and** $S \neq \emptyset$ **do**
**19**  $\quad\quad\quad\quad$ iter $\leftarrow$ S.pop().load()
**20**  $\quad\quad\quad\quad$ $\mu_c \leftarrow$ iter.getNext()
**21**  $\quad\quad\quad$ **if** $\mu_c = nil$ **then return** nil
**22**  $\quad\quad$ **non interruptible**
**23**  $\quad\quad\quad$ **if** *iter* $\neq$ *nil* **then**
**24**  $\quad\quad\quad\quad$ S.push(iter.save())
**25**  $\quad\quad\quad\quad$ iter $\leftarrow$ nil
**26**  $\quad\quad\quad$ n $\leftarrow \mu_c[?o]$
**27**  $\quad\quad\quad$ **if** $\exists (n, d) \in V$, $d \leq |S|$ **then**
**28**  $\quad\quad\quad\quad$ **continue**
**29**  $\quad\quad\quad$ V.add($(n, |S|)$)
**30**  $\quad\quad\quad$ **if** $|S| < MaxDepth$ **then**
**31**  $\quad\quad\quad\quad$ child $\leftarrow$ createIter($n$, $p$, ?$o$)
**32**  $\quad\quad\quad\quad$ S.push(child.save())
**33**  $\quad\quad\quad$ CT.add($(tp_{id}, \mu, (n, |S|))$)
**34**  $\quad\quad\quad$ solution $\leftarrow \mu \cup \mu_c$ ; $\mu_c \leftarrow$ nil
**35**  $\quad\quad\quad$ **return** solution
**36**  $\quad$ **return** nil

To expand a node $v$, $pPTC$ creates an iterator $iter = createIter(v, p, ?o)$. Each time $iter.GetNext()$ is called, it returns a solution mapping $\mu_c$ where $\mu_c[?o]$ is the next node reachable from $v$ through a path that conforms to $p$. In Figure 3 the first time the $GetNext()$ operation of the $pPTC$ iterator is called, it expands the node $T4$. Expanding $T4$ with $p = Sc$ is equivalent to evaluate the triple pattern `T4 Sc ?z`. Consequently, $pPTC$ calls the function $createIter(T4, Sc, ?z)$ to create a *ScanIterator* on the top of the stack $S$ and calls its $GetNext()$ operation to retrieve the first child of $T4$, i.e., $T2$. When $pPTC$ want to expand $T2$, the iterator used to expand $T4$ is saved and a new iterator is created at the top of the stack. As depicted in Figure 3, compared to a traditional DLS like Algorithm 1 (Lines 16-18) the siblings of $T2$ are not stored on the stack before expanding $T2$. Because a preemptable iterator is used to explore $T4$, it can be resumed later to continue the exploration of $T2$ siblings, i.e., only iterators need to be saved, one for each node on the current path. As the space complexity of a saved preemptable iterator is bounded by the size of the query plan and not the size of the data [10], by limiting the exploration depth at $k$, we ensure that the size of $S$ is bounded by $k$. Consequently, the $pPTC$ iterator can be saved and resumed in $O(k)$.

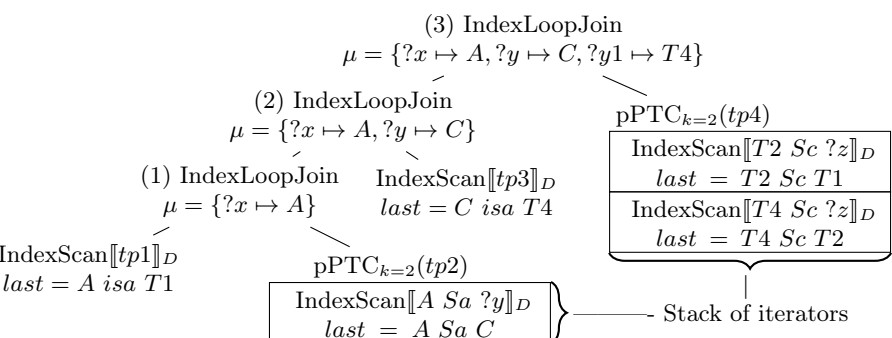

Fig. 3: Physical execution plan of query $Q3$ with the internal state of iterators for $k = 2$

**Quanta and complexities** During one quantum, the $pPTC$ operator maintains a structure to keep track of visited nodes with their corresponding depth. To be preemptable, this structure has to be flushed at the end of the quantum, keeping only the stack of iterators between two quanta. In the worst case, we can consider visited nodes as always empty. In this case, the $pPTC$ iterator enumerates all simple paths leading to a $\#P$ complexity as described in [2]. In the best case, the $pPTC$ iterator has the same time complexity as $PTC$.

**Controls for the PTC-client** To allow a smart client to resume a property path query and continue the evaluation beyond the frontier nodes, visited nodes are contextualized and sent to the client. For each visited node, a $pPTC$ iterator generates a *control tuple* $ct = (tp_{id}, \mu, (n, d))$ (Line 33) where $tp_{id}$ is the identifier of the path pattern that produced $ct$, $\mu$ is the current mappings when $ct$ has been produced and $(n, d)$ is a pair representing a visited node with its depth. For example, suppose that the server is processing query $Q3$ using the physical query plan of Figure 3. When the first index loop join is activated, it pulls the bag of mappings { $?x \mapsto A$ } from its left child, and next calls the $GetNext()$ operation of the $pPTC$ iterator, i.e., its right child. In this context, the $pPTC$ iterator explores node $A$ at depth 1 by evaluating `A Sa ?y`, returns the bag of mappings { $?x \mapsto A, ?y \mapsto C$ } and generates the control tuple $(TP2_{id}, \{?x \mapsto A\}, (A, 1))$.

All control tuples generated during a quantum are stored in a shared memory which is specific to the query during a quantum. At the end of the quantum, control tuples and solution mappings are sent to the client. The data transfer of the control tuples represents the overhead of our $PTC$ approach. To reduce the data transfer, control tuples $ct_1, ..., ct_n$ that share the same $tp_{id}$ and $\mu$ are grouped together into a tuple $(tp_{id}, \mu, [(n_1, d_1), ..., (n_n, d_n)])$.

---
**Algorithm 3:** PTC-Client
---

**Data:**
$MaxDepth$: depth limit
$FIFO$: empty queue of tuples (query $Q$, $ptc_{id}$, frontier node $n$)
$V$: maps each $ptc_{id}$ to a set of pairs (node, depth)
$R$: empty multi-set of solution mappings

**1** **Function** *EvalClient(query)*:
**2**    FIFO.enqueue($(query, nil, nil)$)
**3**    **while** $FIFO \neq \emptyset$ **do**
**4**      $(Q, ptc_{id}, n) \leftarrow$ FIFO.dequeue()
**5**      **if** $\exists (n', d') \in V[ptc_{id}],\ n' = n \wedge d' < MaxDepth$ **then** **continue**
**6**      $(\omega, ct) \leftarrow$ ServerEval($Q$)
**7**      $R \leftarrow R \cup \omega$
**8**      **for** $(tp_{id}, \mu, vc) \in ct$ **do**
**9**        $ptc'_{id} \leftarrow$ hash($\mu, tp_{id}$)
**10**       **for** $(node, depth) \in vc$ **do**
**11**        **if** $\exists (n', d') \in V[ptc'_{id}],\ n' = node$ **then**
**12**          $V[ptc'_{id}]$.add($node$, min($d'$, $depth$))
**13**        **else**
**14**          $V[ptc'_{id}]$.add($node, depth$)
**15**          **if** $depth = MaxDepth$ **then**
**16**            $Q_e \leftarrow$ ExpandQuery($Q, tp_{id}, \mu, node$)
**17**            FIFO.enqueue($Q_e, ptc'_{id}, node$)

**18**    **return** R

---

## 4.3 The PTC-Client

The general idea of the PTC-Client is to use the control tuples returned by the server-side *pPTC* iterators to expand frontier nodes until no more frontier nodes can be discovered, i.e., transitive closures are complete.

Algorithm 3 describes the behavior of the PTC-client. It is fundamentally an iterative Breadth-First Search (BFS) algorithm that traverses frontier nodes. The FIFO queue stores the frontier nodes to traverse with their context. $R$ is the multi-set of results. The $V$ variable represents the visited nodes. As a path expression may be instantiated many times, we store a set of visited per instance of path expression, i.e., $ptc_{id}$.

Figure 4 illustrates the first iteration of Algorithm 3 using query $Q3$ of Figure 2b. First, the query $Q3$ is evaluated on the server by calling *ServerEval* (Line 6). *ServerEval* accepts any SPARQL property path query and returns a set $\omega$ of solution mappings and a set $ct$ of control tuples. The sets $\omega$ and $ct$ for query $Q3$ are depicted in Figure 4 by the two tables. As we can, all visited nodes are discovered with a depth = 1, as MaxDeph = 1, they are all frontier nodes. Consequently, the Algorithm 3 will expand $Q3$ with all these frontiers nodes.

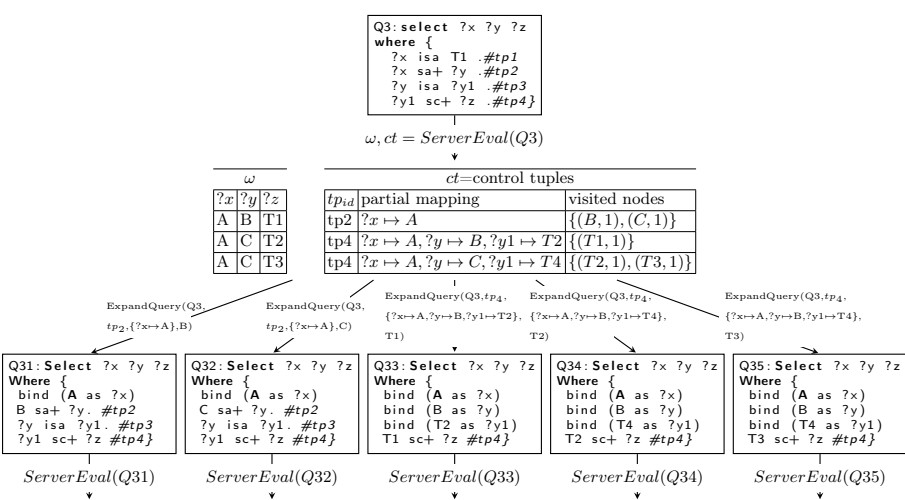

Fig. 4: First iteration of ClientEval($Q3$) as defined in Algorithm 3 with graph $D$ and $MaxDepth = 1$

*ExpandQuery* takes a query $Q$, a set of partial mappings and a frontier node $n$ as input (Line 16) and produces a new query as output. *ExpandQuery* processes in three steps. (1) The subject of the path pattern identified by $tp_{id}$ in $Q$ is replaced by $n$. (2) Triple patterns $tp$ in $Q$ such as $\mu(tp)$ is fully bounded are removed. (3) To preserve the mappings $\mu$ from $Q$ to $Q_e$, a BIND clause is created for each variable in $dom(\mu)$.

Figure 4 illustrates the query returned by *ExpandQuery* for each frontier node returned by *ServerEval*($Q3$). Queries $Q31$, $Q32$, $Q33$, $Q34$ and $Q35$ are finally pushed in the FIFO queue (Line 17) to be evaluated at the next iteration. It could happen that the evaluation of an expanded query reached an enqueued frontier node with a shortest path (Line 12). In this case, the expanded query is not evaluated (Line 5).

## 5 Experimental Study

In this experimental study, we want to empirically answer the following questions: What is the impact of the $maxDepth$ and the time quantum parameters on the evaluation of SPARQL property path queries, both in terms of data transfer, number of HTTP calls and query execution time? How does the PTC approach perform compared to smart client approaches and SPARQL endpoints?

In our experiments *Jena-Fuseki* and *Virtuoso* are used as the baselines to compare our approach with SPARQL endpoints, while the multi-predicate automaton approach [1] is used as the baseline for the smart client approaches. We implemented the PTC operator in Python as an extension of the SAGE server, while the PTC client is implemented in JavaScript. The resulting system, i.e., the SAGE server with our PTC operator and the JavaScript client, is called

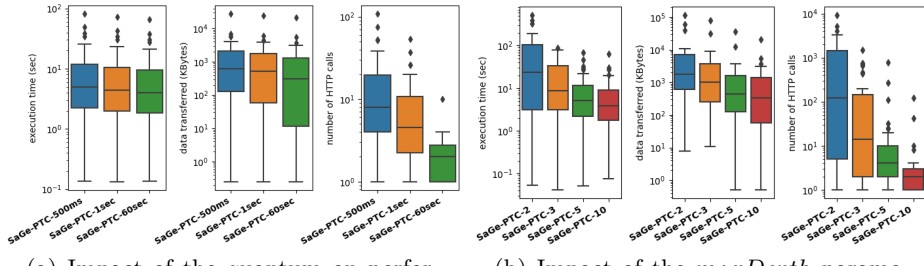

(a) Impact of the quantum on performance

(b) Impact of the *maxDepth* parameter on performance

Fig. 5: Impact of the different parameters on performance for the gMark queries

*SaGe-PTC*. The code and the experimental setup are available on the companion website [3].

### 5.1 Experimental setup

***Dataset and Queries:*** The dataset and queries are generated by gMark [3], a framework designed to generate synthetic graph instances coupled with complex property path query workloads. We use the "Shop" use-case configuration file[4] to generate a graph instance of 7,533,145 triples and a workload of 30 queries. All our queries contain from 1 to 4 transitive closure expressions, for which numerical occurrences indicators have been replaced by Kleene plus $"+"$ operators.

***Compared Approaches:*** We compare the following approaches:

- *SaGe-PTC* is our implementation of the PTC approach. The dataset generated by gMark is stored using the SaGe HDT backend. The SaGe server is configured with a page size limit of 10000 solution mappings and 10000 control tuples. Different configurations of *SaGe-PTC* are used. (i) *SaGe-PTC-2*, *SaGe-PTC-3*, *SaGe-PTC-5*, *SaGe-PTC-10* and *SaGe-PTC-20* are configured with a time quantum of 60 seconds and a *maxDepth* of 2, 3, 5, 10 and 20, respectively. (ii) *SaGe-PTC-500ms*, *SaGe-PTC-1sec* and *SaGe-PTC-60sec* are configured with a *maxDepth* of 20 and a time quantum of 500ms, 1sec and 60sec, respectively.

- *SaGe-Multi* is our baseline for the smart client approaches. Property path queries are evaluated on a SaGe smart client using the decomposition approach defined in [1]. For a fair evaluation, *SaGe-Multi* runs against the *SaGe-PTC* server with a time quantum of 60 seconds. We did not include Comunica [15] in the setup as it has already been compared with *SaGe-Multi* in [1] and *SaGe-Multi* dominates Comunica for all evaluation metrics.

- *Virtuoso* is the Virtuoso SPARQL endpoint (v7.2.5) as of December 2020. *Virtuoso* is configured **without quotas** in order to deliver complete results.

---

[3] https://github.com/JulienDavat/property-paths-experiments
[4] https://github.com/gbagan/gmark/blob/master/use-cases/shop.xml

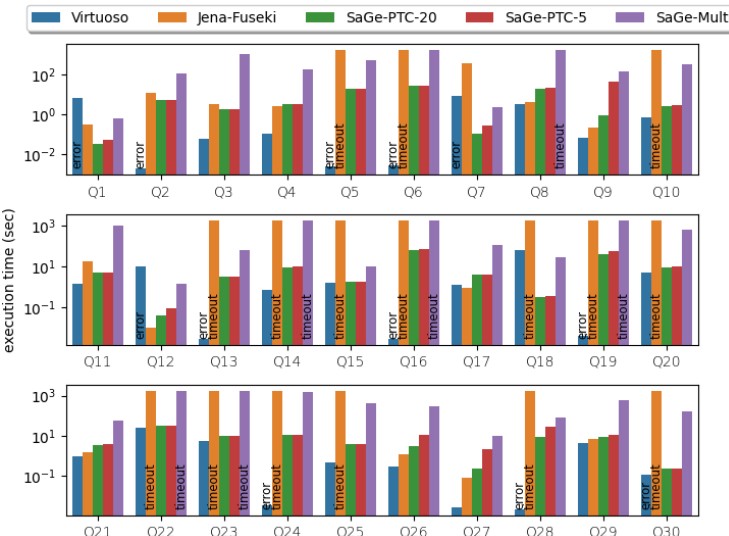

Fig. 6: Execution time per query for SPARQL endpoint and smart client approaches compared to our PTC approach

We also configured *Virtuoso* with a single thread to fairly compare with other engines.

– *Jena-Fuseki* is the Apache Jena Fuseki endpoint (v3.17.0) with the same configuration as *Virtuoso*, i.e., without quotas and a single thread.

**Evaluation Metrics:** Presented results correspond to the average obtained of three successive executions of the queries workload. Each query is evaluated with a time-out of 30 minutes. (1) *Execution time* is the total time between starting the query execution and the production of the final results by the client. (2) *Data transfer* is the total number of bytes transferred to the client during the query execution. (3) *Number of HTTP calls* is the total number of HTTP calls issued by the client during the query execution

**Hardware Setup:** We run our experimentations on a google cloud virtual machine (VM) instance. The VM is a c2-standard-4 machine with 4 virtual CPU, 16GB of RAM and a 256GB SSD. Both clients and servers run on the same machine. Each client is instrumented to count the number of HTTP requests sent to the server and the size of the data transferred to the client.

## 5.2 Experimental results

*What is the impact of the quantum on performance ?* To measure the impact of the quantum on performance, we run our workload with different quanta; 500ms, 1sec and 60sec. The *maxDepth* is set to 20 for each quantum, such as all queries terminate without frontier nodes. Figure 5a presents *SaGe-PTC* performance for *SaGe-PTC-500ms*, *SaGe-PTC-1sec* and *SaGe-PTC-60sec*.

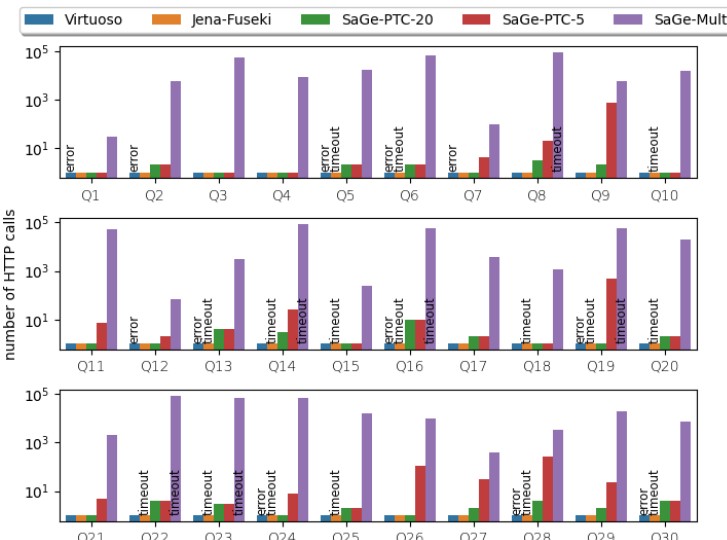

Fig. 7: Number of HTTP calls per query for SPARQL endpoint and smart client approaches compared to our PTC approach

As expected, increasing the quantum improves performance. With large quantum, a query needs less calls to complete, which in turn improves the execution time. Concerning the data transfer, the visited nodes of *pPTC* iterators are flushed at the end of each quantum, which leads to revisit already visited nodes. Consequently, the less a query needs quanta to complete, the less it transfers duplicates.

*What is the impact of maxDepth on performance ?* To measure the impact of *maxDepth* on performance, we run our 30 queries with different values of *maxDepth*; 2, 3, 5 and 10. To reduce the impact of the quantum, we choose a large quantum of 60 seconds. Figure 5b presents *SaGe-PTC* performance for *SaGe-PTC-2*, *SaGe-PTC-3*, *SaGe-PTC-5* and *SaGe-PTC-10*.

As we can see, the *maxDepth* impacts significantly the performance in terms of execution time, data transfer and number of calls. Increasing the *maxDepth* drastically improves performance because it allows to capture larger transitive closures. This means less control tuples are transferred to the client and less expanded queries are executed on the server.

*How does the PTC approach perform compared to the smart client approaches and SPARQL endpoints ?* The PTC approach computes SPARQL property path queries without joins on the client. When *maxDepth* is high enough, no expanded queries are sent to the server. We just have to pay the web preemption overheads and the duplicates transferred by the PTC approach. Consequently, we expect our approach to be somewhere between SPARQL endpoints and smart clients in terms of performance. Close to SPARQL endpoints when *maxDepth* is high and better than smart clients in the general case. We compare *SaGe-PTC* with *SaGe-*

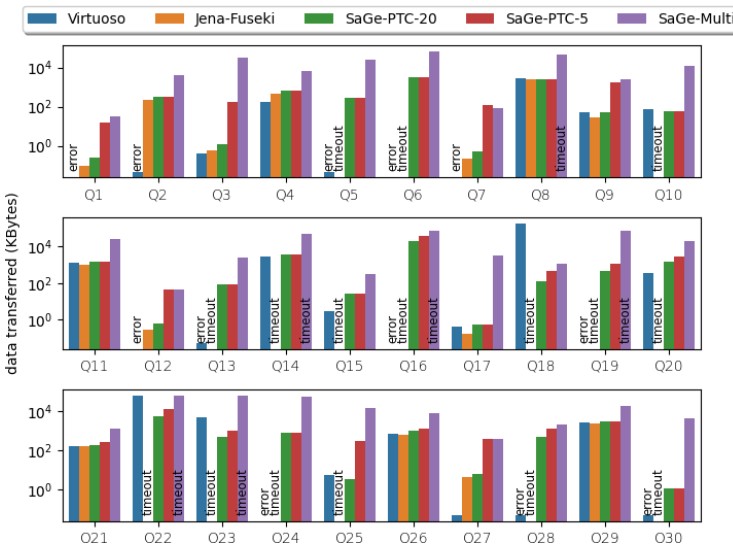

Fig. 8: Data transfer per query for SPARQL endpoint and smart client approaches compared to our PTC approach

*Multi*, *Jena-Fuseki* and *Virtuoso*. We run *SaGe-PTC* with both $maxDepth = 5$ (*SaGe-PTC-5*) and $maxDepth = 20$ (*SaGe-PTC-20*). Figures 6, 8 and 7 respectively present the execution time, the data transfer and the number of HTTP calls for each approach.

As expected, *SaGe-PTC* outperforms *SaGe-Multi* regardless the query. Because *SaGe-PTC* does not decompose BGPs, the number of calls is drastically reduced, as shown in Figure 7. Concerning the data transfer, both *SaGe-PTC* and *SaGe-Multi* transfer the visited nodes. However *SaGe-PTC* does not transfer any intermediate results, saving a lot of data transfer.

Compared to *Jena-Fuseki* both approaches use a similar graph traversal algorithm. However *Jena-Fuseki* has no overheads, i.e., is optimal in terms of data transfer and number of calls. Only one call is sent per query and only the final results are transferred to the client. Consequently, we expect *Jena-Fuseki* to perform better than *SaGe-PTC*. Surprisingly *Jena-Fuseki* does not dominate the PTC approach. *SaGe-PTC-20* is very close to *Jena-Fuseki* for queries where *Jena-Fuseki* does not time-out. As queries return no frontier nodes, overheads compared to *Jena-Fuseki* are small. The differences between the two approaches are mainly due to a join ordering issue. Compared to *SaGe-PTC-20*, with *SaGe-PTC-5* queries need to send expanded queries to terminate. As expected, *SaGe-PTC-5* offers performance between that of SPARQL endpoints and that of smart clients. For most queries, its performance are very close to *SaGe-PTC-20*. We conjecture that most transitive closures for our queries can be computed with small *maxDepth*. When there is a large number of frontier nodes to explore, performance degrades but remains better than those of smart client approaches.

We expect *Virtuoso* to be optimal as it implements the state of art query optimization techniques. Surprisingly, *Virtuoso* does not dominate *SaGe-PTC*. *Virtuoso* generates errors for 12 queries out of 30. It cannot execute the 12 queries either because of the missing of a starting point or because it has not enough space resources to materialize the transitive closure. *Virtuoso* issues are mainly due to the simple path semantics when dealing with dense graphs. Of course, the number of calls for *Virtuoso* is optimal. However, using a simple path semantics leads to high data transfer when path queries are executed against dense graphs.

## 6 Conclusion

In this paper, we proposed an original approach to process SPARQL property path queries online and get complete results. Thanks to a preemptable Partial Transitive Closure operator, a smart client is ensured to grab all mappings that are reachable at a depth fixed by the server. Thanks to control information delivered during SPARQL property path queries processing, a smart client generates queries to find missing mappings. Unlike current smart clients, the PTC smart client does not break BGPs containing path patterns. Even in presence of path patterns, all joins are performed on server-side without the need to transfer intermediate results to the client. As demonstrated in the experimentations, the PTC approach outperforms existing smart clients and reduces significantly the gap of performance with SPARQL endpoints. This approach raises several interesting perspectives. First, there is a large room for optimisation: better join ordering in presence of path patterns, pruning some calls when path patterns are "reachability" oriented, and better evaluation of resuming queries according to their shapes. Second, it may be interesting to explore if partial transitive closure is compatible with partial aggregates [7]. If the aggregation functions are computed on client-side, then there is no issue for computing aggregation in presence of path patterns. However, partial aggregates computed with partial transitive closures on server-side may return incorrect results.

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

**Acknowledgments** This work is supported by the ANR DeKaloG (Decentralized Knowledge Graphs) project, ANR-19-CE23-0014, CE23 - Intelligence artificielle.
