# OpenReview forum: "Processing SPARQL Property Path Queries Online with Web Preemption"
_eswc-conferences.org/ESWC/2021/Conference/Research_Track — ESWC 2021 Research_

### Official Review · AnonReviewer3 · 2021-01-03
**solid piece of work, but some aspects remain unclear**

**Rating:** 2
**Confidence:** 5
**Impact:** 3
**Design And Technical Quality:** 4

**Review:**

## After Rebuttal

I am happy with the authors' response. They have addressed all my concerns and answered all the questions to my satisfaction. I am certain that the mentioned changes to the paper can and will be done in the context of preparing the camera-ready version and that these changes will greatly improve the quality of paper and the clarity of the presented approach. With this, I am comfortable to raise my score to 'Accept'.

## Original Review

The paper introduces a novel approach to support property path patterns in SPARQL servers that can preempt query execution requests. In contrast to existing approaches, the presented approach evaluates property path patterns completely on the server side, in combination with the other patterns of the given query. Hence, by using this approach, there is no need for a client-side algorithm that decomposes the evaluation into multiple processing steps that requires back and forth between the client and the server (other than the interactions caused by preemption of query executions). The experimental evaluation shows the superiority of the approach in terms of query execution times, data transfer, and number of requests.

I really enjoyed reading this paper! It is very well written. There is a clear motivation and a clear problem statement, the approach is presented in sufficient detail (with one exception, see below), a running example accompanies the definition of the approach, and the evaluation provides insights about all major aspects of the approach.

The approach itself is a refreshing and original adaptation of an existing graph traversal algorithm. My only major complaint is the following: While the main algorithms for the server side and for the client side are presented in detail, respectively, the interaction between these algorithms remains somewhat unclear (details below). Due to this issue, I do not feel comfortable to propose an accept for the paper as is. However, I hope that the authors can help clarify my remaining confusions during the rebuttal.

In the remainder of this review I will first elaborate more on the aforementioned issue and, thereafter, list a few minor issues and suggestions to address these issues.


### Interaction between client and server algorithms unclear:

The main problem is that the function 'serverEval' in line 9 of Algorithm 3 is not described. While I can guess based on its name and based on how the function is used within the algorithm, I still wonder:

Q1: Is it correct to assume that the input to this function is a complete SPARQL query?

Q2: What exactly does this function do?

Q3: Does this function really return only a single solution mapping together with a control tuple?

Q4: If the answer to Q3 is no, then what kind of a thing is $\mu$ and how is the returned control tuple related to this thing? (and I strongly suggest to use a different symbol because $\mu$ is usually used for solution mappings in the context of SPARQL)

Q5: If the answer to Q3 is indeed yes, what happens to the other solution mappings that are solutions for the query passed to 'serverEval' in line 9? When are these solutions processed?

Q6: Also, if the answer to Q3 is yes, how is the returned control tuple related to the return solution mapping? In particular, if the query passed to 'serverEval' contains multiple property path patterns and, thus, multiple pPTC iterators in the server-side plan, the control tuple of which of these pPTC iterators is returned here?

The latter question also reveals that some details about the server-side processing are missing: I understand that, during the query execution within the server, the GetNext() function of the pPTC iterator (Algo.2) returns one solution mapping at a time, together with the corresponding control tuple. Each such solution mapping is a solution for the part of the query that is covered by the subtree of the query plan rooted in the respective pPTC iterator. These solution mappings are then consumed by parent iterators that are higher up in the query plan, which may extend these solution mappings with additional variable bindings. This makes me wonder:

Q7: How are the control tuples maintained in this context? More precisely, if the parent iterator is responsible for a standard triple pattern, what happens to the control tuples of the solution mappings that it consumes? And if a parent iterator is another instance of the pPTC iterator (for another property path pattern in the query), does this instance extend the control tuples (how?) or create new control tuples separately (if so, what happens to the original ones?)?

Some more questions about the client-side algorithm (Algorithm 3):

Q8: What exactly does the 'expand' function in line 19 do? There is no description of this function in the text!

Q9: What is the actual result of this algorithm? Does it return the query result? There is no return statement.


### Minor issues:

I1: There seems to be an inconsistency in the description of the PTC operator (first paragraph of Sec.4.1) and the corresponding example (third paragraph): The path expression in the example is 'Sa' and the result includes nodes D and E. According to the description of the operator, there must "exist[] a path that conforms to" the given path expression. However, there is no path from A to D or to E that conforms to the path expression 'Sa' (at least, not by my understanding of what it would mean for a path to "conform" to a path expression). Instead, there are paths from A to D and from A to E that conform to the path expression 'Sa+' (or to 'Sa/Sa'). I think the description of the operator needs to be fixed to make explicit that "conformance" involves transitivity.

I2: Related to the previous point, the word "length" in the description is incorrect. What the authors mean to denote by $d$ is the number of sub-paths of the "conforming" path that match the given path expression. For path expressions that consist of a single step that can be matched by a single edge (as in the example), this number coincides with the actual length of the "conforming" path, but if the path expression matches sub-paths of multiple edges (i.e., sub-paths of length greater than 1), then the length of the "conforming" path is different from its value $d$.

I3: I assume that the input argument 'path' for the 'ALP' functions in Algorithm 1 is not allowed to contain recursive traversal operators (* and +). While there is a related comment for the approach in general, I suggest to make this restriction explicit again in the description of the algorithm (or, at least, in Algorithm 1 itself).

I4: In the description of the pPTC iterator in Sec.4.2, I suggest to add 1-2 sentences that emphasize explicitly that the size of the stack $S$ is bounded by MaxDepth and, due to this upper bound, the operations 'Save' and 'Load' can be done in constant time. There is some more general mention of this fact earlier in the paper, but here is the point to reiterate on this fact again and to pinpoint the concrete reason/justification for it.

I5: The last paragraph of page 7 talks about "outer join" which is incorrect. There is no outer join here (the query does not contain an OPTIONAL). I propose to say: "...from its child iterator"

I6: In the same sentence: "...of the pPTC iterator." To avoid any confusion, it should be made explicit which of the two pPTC iterators is meant here.

I7: In line 17, Algorithm 2 uses a 'hasNext()' operation of the iterator. This operation is undefined. This issue can be fixed as follows: Given that the 'GetNext()' operation may return 'nil', which I assume is meant to capture cases in which there is no next solution mapping to be returned, I propose to replace lines 17-18 as follows.

   line 17:  $\mu_c$ = iter.GetNext()

   line 18:  if $\mu_c \neq$ nil then

I8: Another inconsistency in Algorithm 2 is that line 18 (or line 17 according to the proposal in my previous point) retrieves only a solution mapping from the 'GetNext()' operation of the iterator while this operation actually returns a pair consisting of a solution mapping and a corresponding control tuple (see line 28 in the algorithm).

I9: Yet another thing that needs to be clarified is the meaning of the 'val' function used in line 22 of the algorithm.

I10: Similar small inconsistencies and mistakes exist in Algorithm 3: First, according to line 4, R is a "set of term[s]" (which, I assume, is meant to be RDF terms). However, line 10 performs a union of this set with a solution mapping. What does this mean?

I11: Second, I am not sure whether it is correct that the first input argument in line 19 should really be Q (the input of the 'eval' function itself); I would assume that this is meant to be the variable 'query' instead (which was assigned in line 6).

I12: Third, in line 11, it should be $(id,\mu,vc)$ instead of $(\mu,id,vc)$.

I13: Additionally, the part around lines 14-15 can be simplified: That is, adding to V[key] in line 15 is necessary only if min(depth,d)=depth.

I14: Something is messed up in the text of the second bullet point on page 12.

I15: In the same bullet point, once it says "SAGE smart client" and once it says "SAGE client" -- assuming that this is meant to refer to the same client, I propose to be consistent and use only one of these two terms.

I16: The first sentence in Sec.5.2 talks about "workloads" (plural!). According to my understanding of the experimental setup, there was only a single query workload. This needs to be clarified.

I17: In the next sentence, it needs to be clarified whether the 30 mins timeout is per query or per workload.

**Anonymity:**

No, I would like my review to be deanonymized.

**Reuse And Availability:**

4: High

**Strong Points:**

S1: The presented approach is original and advances the state of the art in the corresponding line of work (about SPARQL servers that can preempt query execution requests).

S2: The paper is very well written and easy to understand (with the exception of the issue mentioned in my review).

S3: The evaluation provides insights about all major aspects of the approach.

**Subreviewer:**

I submitted this review.

**Weak Points:**

W1: Some aspects of the presented approach remain somewhat unclear (as detailed in my review).

W2: While the paper claims that "the code and the experimental setup are available [at] [https://github.com/JulienDavat/property-paths-experiments](https://github.com/JulienDavat/property-paths-experiments)", there does not exist any github repo at the given address.

---

> ### Author Rebuttal · Authors · 2021-01-29
>
> We thank the reviewer for their constructive comments.
>
> Q1:
> * Yes, the input of the serverEval function is a complete SPARQL query.  Figure 4 illustrates the first iteration of the “while loop” of the function eval(Q3) of Algorithm 3. To avoid confusion, we replaced all ‘eval’ of Figure 4 with ‘serverEval’ and the appropriate SPARQL query as the input parameter, e.g., serverEval(Q3). We also replaced the label of Figure 4 to be more explicit. The updated  Figure 4 is available on https://github.com/JulienDavat/property-paths-experiments.
>
> Q2:
> * serverEval(Q) is a client-side function that manages the execution of the query Q on the server until query termination. For instance, if Q3 in Figure 4 requires several quanta to complete, serverEval will resend Q3 until completion with maxDepth=1.
>
> Q3:
> * No it doesn’t. The function serverEval returns two sets; a set of solution mappings and a set of control tuples. For instance, Figure 4 describes the output of serverEval(Q3). As we can see in Figure 4, once Q3 completes, serverEval returns two sets represented by two tables. The table on the left corresponds to the set of solution mappings and the table on the right corresponds to the set of control tuples. To avoid confusion, we put headers on tables in Figure 4.
>
> Q4:
> * We replaced \mu with  \omega
>
> Q5:
> * The answer to Q3 was no.
>
> Q6:
> * The set of solution mappings and control tuples are 2 independent sets. The set of control tuples may contain information returned by more than one pPTC operator.
>
> Q7:
> * Compared to the solution mappings, control tuples are not propagated in the pipeline of iterators (to be consistent with the definition of an iterator). When the GetNext function of our pPTC iterator generates a control tuple, it is stored in a “shared” memory  relative to the query during the quantum. At the end of the quantum, the set of solutions mappings and control tuples are sent to the client. We added this to the explanation of algorithm 2.
>
> Q8:
> * We would like to apologize that the explanations of  the “expand”  function in page 9 :“We expand Q3 with the partial mappings of the visited node B…” are not sufficient. Expand(Q, mu,id, frontier) generates the query Q’ that allows to continue the execution of Q from the frontier node. In Figure 4, Expand(Q3, { ?x->A }, tp2, B) rewrites Q3 as the leftmost query.    As Q3 hits the frontier at node B with ?x mapped to A. Therefore “Expand” rewrites Q3 by first binding ?x with A,  and replaces ?x with B in the path pattern identified by tp2. All leaf queries of Figure 4 are generated following the same rules. We added more detailed explanations in the paper.
> Q9:
> * The overall results of the algorithm are the union of the results of all queries evaluated by the algorithm. We added the return statement to the paper.
>
> I1:
> * In the description of PTC(n,p,k), p is the path expression on which the Kleene star or plus are applied. For instance, if we have the path expression “Sa+”, p will equal to  “Sa”. We clarified the description of PTC.
>
> I2:
> *  “length” is the number of sub-paths of the “conforming” path.  We agree, we clarified this in the paper.
>
> I3:
> * Yes, currently recursive traversal operators are not supported. We  added this restriction to Algorithm 1.
>
> I4:
> * We agree. We added sentences to emphasize explicitly that the size of the stack  S is bounded by MaxDepth and, due to this upper bound, the operations 'Save' and 'Load' can be done in constant time.
>
> I5:
> * We agree. We replaced “ from its outer join” by “from its left child iterator”.
>
> I6:
> * We agree. We specified explicitly which pPCT iterator we considered
>
> I7:
> * We agree. We updated our algorithm to take into account the reviewer suggestion
>
> I8:
> * The control tuple must not be returned by the GetNext() function. Compared to the solution mappings, control tuples are not propagated in the pipeline of iterators (to be consistent with the definition of an iterator). When the GetNext function of our pPTC iterator generates a control tuple, it is stored in a “shared” memory relative to the query during the quantum. At the end of the quantum, the set of solutions mappings and control tuples are sent to the client.  We clarified this.
>
> I9:
> * The val function returns the value of the variable that corresponds to the last visited node. We clarified this.
>
> I10:
> * Yes, R is in fact a set of solution mappings and not a set of RDF terms. We updated algorithm 3.
>
> I11:
> * We agree, it is the “query” variable that must be passed to the “expand” function. We updated algorithm 3.
>
> I12:
> * We updated algorithm 3
>
> I13:
> * this is a possible optimization.
>
> I14:
> * there is a layout problem here. We fixed this.
>
> I15:
> * We used “SAGE smart client" everywhere.
>
> I16:
> * There is only one workload. We fixed this.
>
> I17:
> * The 30 minutes timeout is per query. We fixed this.
>
> W2:  github repo at the given address.
> * The repository is up now  https://github.com/JulienDavat/property-paths-experiments

---

### Official Review · AnonReviewer1 · 2021-01-06
**Promising work let down by poor writing**

**Confidence:** 4
**Impact:** 4
**Design And Technical Quality:** 3

**Review:**

This paper studies the problem of evaluating (simplified) property paths expressions in a preemptive way, that is, allowing them to be paused and later resumed, without needing to transfer too much data between the server and the user. The idea behind this is to introduce a partial transitive closure operator which computes the paths iteratively, and can be paused, in which case it returns the nodes currently reached (similarly as in any graph search algorithm), or stores them locally. The idea is rather elegant, and is implemented using depth limited search (i.e. DFS up to a pre-specified depth). The authors present a pseudo-code for their approach, and an extensive experimental evaluation.

While I find the topic interesting, and the proposed approach very promising, I believe that the writing of the paper is rather poor, and fails to covey the underlying ideas very effectively. The authors also do not do a very good job to differentiate this paper from their own previous work, and at the time of writing the review, the source code and the experimental setup were not available.

That being said, the main idea behind this work is quite nice, and actually has a lot of potential to work in practice. Should the source code of this paper be made available during the rebuttal, and if the writing marginally improves (I do not think major improvements possible  for the camera ready version), I might actually be inclined to support the paper, but as it stands, it is difficult to do so (which I find a real shame, due to the underlying idea being quite nice).


**Anonymity:**

Yes, I would like my review to remain anonymous.

**Rating:**

-1: Weak Reject

**Reuse And Availability:**

1: Very low

**Strong Points:**

- An interesting, and highly relevant problem.
- A promising approach which seems reasonably efficient (one can always construct pathological cases, but this is true for any database algorithm).
- Presentation, although poorly realized, actually point towards how this approach would be implemented.

**Subreviewer:**

I submitted this review.

**Weak Points:**

- The writing is a mess honestly. I will provide several examples of where this could be improved next.
- First, the differentiation from the author's QuWeDa 2020 paper is not done at all. Given that the initial paragraphs read identically, I initially thought that I was reading the same paper. More on how this paper builds on top of the previous one should be said.
- Next, in Section 4 the algorithms appear far away from where they are actually discussed, making it difficult to follow what is going on. Furthermore, the authors do a very poor job of being specific about how the transitive operations work, and what do they actually compute. For instance, while Algorithm 1 is conceptually correct, one has to read quite further ahead to actually realize that "path" refers to a non-transitive portion of a property path, over which the Kleene star is applied. As written now, in particular in line 16 of the rightmost ALP, this is completely hidden. This also assumes that the property path is decomposed in a way that the proposed algorithm can process it, which is definitely doable, but should be explained properly.
- The same displacement of figures is present in Section 5, as is in Section 4.
- The experimental setup is not that well explained. In particular, I did not understand how the data transfer/number of requests is measured. Is it assumed that there is always an endpoint on the google cloud and queries are submitted from a personal machine? Next, the data size/distribution is not explained at all.
- As of the time of writing this review, the link for accessing the experiment data/implementation on github was unavailable.
- Given that the authors admit that repeated answers might be returned due to the DLS implementation, I was surprised that the percentage of such duplicates was not measured in the experiments. This would actually allow to see the overhead, and probably confirm that the approach is rather reasonable.
- I do not understand the PTC guarantee at the bottom of page 4: DLS has length guarantee k, but starting from a single node, there can be many such paths! If one is returned OK, but since many might be, I do not see this as resuming in constant time; actually, reading further one gets the hand of it, but it should be stressed more heavily that the only reason why this is possible is the absence of nested stars!
- On p2, the authors claim that recursive queries are a way of implementing property paths, but do any endpoints actually implement recursive queries? Given that this is not in the standard.

---

> ### Author Rebuttal · Authors · 2021-01-29
>
> We thank the reviewer for their helpful comments.
>
> First, the differentiation from the author's QuWeDa 2020 paper is not done at all. Given that the initial paragraphs read identically, I initially thought that I was reading the same paper. More on how this paper builds on top of the previous one should be said.
>
>  * As this paper addresses a similar issue as [1], the context and motivation are similar. We tried to explain the positioning of this paper compared to [1] in section 2. in [1], the preemptive server does not support transitive closures. Consequently, the smart client breaks the BGPs and decomposes the query into subqueries to isolate the property path. For example, to evaluate the query “SELECT * WHERE { “obama” a ?x . ?x b+ ?y }”, the two gtriple patterns (“obama” a ?x) and (?x b+ ?y) are evaluated separately and then a join is performed on the client to build the final result.  We completely agree that this should be elaborated further. We added this  example in the paper.
>
> Next, …. As written now, in particular in line 16 of the rightmost ALP, this is completely hidden. This also assumes that the property path is decomposed in a way that the proposed algorithm can process it, which is definitely doable, but should be explained properly.
>    * The “path” parameter of the APL function is effectively the path expression over which the Kleene star is applied. Moreover, as our operator only supports transitive path expressions (Kleene stars), non-transitive path expressions, such as sequences and alternatives, must be decomposed following the rewriting rules defined in the SPARQL semantics.  We agree, we clarified this further.
>
> The same displacement of figures is present in Section 5, as is in Section 4.
>   * We improved the placement of figures.
>
> The experimental setup is not that well explained. In particular, I did not understand how the data transfer/number of requests is measured. Is it assumed that there is always an endpoint on the google cloud and queries are submitted from a personal machine?
>    * To run our experiments, we rented a virtual machine on the google cloud platform that runs both the SaGe server with our PTC operator and the javascript client.To measure the data transfer/number of requests, we instrumented the client to count the number of http queries sent to the server and the size of data received from the server.
>
> Next, the data size/distribution is not explained at all.
>    * The paragraph “Dataset and Queries” in section 5.1 gives details about the data and the queries. The data distribution is defined in the gMark “Shop” scenario configuration file.
>
> As of the time of writing this review, the link for accessing the experiment data/implementation on github was unavailable.
>    * We regret this inconvenience.  The repository is up now at: https://github.com/JulienDavat/property-paths-experiments and plots legends are detailed.
>
> Given that the authors admit that repeated answers might be returned due to the DLS implementation, I was surprised that the percentage of such duplicates was not measured in the experiments. This would actually allow to see the overhead, and probably confirm that the approach is rather reasonable.
>    * The impact of duplicates on the data transfer can be observed in Figure 2a (middle plot). As expected, when the quantum increases, the number of duplicates decreases.  We better explained this figure to highlight duplicate issues.
>
> I do not understand the PTC guarantee at the bottom of page 4: DLS has length guarantee k, but starting from a single node, there can be many such paths! If one is returned OK, but since many might be, I do not see this as resuming in constant time; actually, reading further one gets the hand of it, but it should be stressed more heavily that the only reason why this is possible is the absence of nested stars!
>    * Our PTC operator can be saved and resumed in constant time because we work with iterators. The PTC operator uses a stack of iterators instead of a stack of nodes. When working with iterators, sibling nodes can be retrieved thanks to the iterators “GetNext” operation. Consequently, they do not have to be saved in the stack. Using iterators allow us to guarantee that the stack size is at most k. Then, saving or resuming a stack of size k can be done in constant time.
>
> On p2, the authors claim that recursive queries are a way of implementing property paths, but do any endpoints actually implement recursive queries? Given that this is not in the standard.
>    * We apologize for the confusion; recursive queries and graph traversal algorithms are common approaches for computing transitive closures, we removed “implemented in SPARQL endpoints”.

---

> > ### Comment · AnonReviewer1 · 2021-02-01
> > **Post rebuttal**
> >
> > I would like to thank the authors for their replies. While they do clear up some of the questions I had, I still strongly believe that this paper would benefit from a thorough rewriting that would make the presentation more streamlined and easy to follow. The addition of the implementation is greatly appreciated, but submitting it several months afterwards is not an acceptable practice.
> >
> > Given all this, I will stick to my original score.
> >
> > That being said, the work done here is actually quite good, and I would encourage the authors to present it better and submit to another venue.

---

### Official Review · AnonReviewer4 · 2021-01-11
**Review for Processing SPARQL Property Path Queries Online with Web Preemption**

**Rating:** 1
**Confidence:** 4
**Impact:** 4
**Design And Technical Quality:** 4

**Review:**

### After Rebuttal

I thank the authors for the answer. Some of my concerns are handled, however, as me and other reviewers pointed, the writing and the explanations of some parts of the work can be improved. From the answer, I understood that the authors will improve figure 4 (it would have been great to have access to the new figure in this stage), but I think that more work is needed. Thus, I cannot understand until what extent they will improve the readability of the paper, or also to clarify the issues pointed by the reviewers.
Thus, I change my review to 1 (weak accept), but for me is more like a borderline paper; and this is just because of the writing, since their work is solid and they are proposing a smart new technique.

# General comments

The authors propose a way to evaluate online property paths that can compute complete solutions, instead or partial solutions. The approach consists in using a partial transitive closure operator. Although the work is interesting and the results are good, I think that the paper is hard to read and also their revision of the related work can be considerably improved.

# Detailed comments

The authors start with a motivation of the problem, explaining why the quanta and the maximum number of results can be a problem at the time of evaluating a property path query. Although the problem is well motivated, the introduction is not fluid and the related work has room for improvement. For instance, there is not discussion of some proposal such as the discussed in "Evaluating Navigational RDF Queries over the Web" (Baier, Daroch, Reutter & Vrgoc, 2017). Also the authors claim that some strategies used by SPARQL endpoints rely on Recursive Queries; it would be great to have a detailed list of the strategies used by the most famous engines (Jena, Virtuoso, Blazegraph). As far as I know, none of them uses Recursive Queries.

Also, at some point the authors claim that nested regular expressions (such as (ab*)*) are rare and not of interest. I disagree with this opinion, since use cases such as those studied in "Relative Expressiveness of Nested Regular Expressions" (Barceló, Pérez & Reutter, 2012) and some other papers show that these expressions are in fact important.

With respect to the description of Web Preemption, I think that the description is ok. Also I think that the discussion of the evaluation of property paths and the problems generated with Web Preemption are ok. However, I don't understand the point of declaring the problem at the end of such section.

In section 4, the authors present their algorithms to solve property paths queries with web preemption. However I think that this section is hard to read and can be improved. For instance, there are some examples, but it would be great to have an example that connects all the parts of the strategy to understand the complete workflow of the approach. Also, there are practical issues that are not discussed. When a user asks for a property path query to an endpoint, what happens when the quanta is over? The user receives some partial results or he has to wait for the endpoint to have all the results? Also, I would like to know how long are the times between distinct quanta in practice.

With respect to the queues of remaining nodes. They are stored at the server or at the client? How is the interaction between client and server? This can be explained better. Also, please clarify the figure 4, it is not useful to understand your approach. Moreover, it would be great to discuss what happens in case of concurrent queries done by distinct users.

Then, regarding the experimental section, I wanted to take a look to the implementation, but the link was broken. it would be great to have access to the queries generated by the benchmark, as sometimes the queries created by GMark does not make sense. I think that this section is difficult to read, particularly, the names for distinct implementations are very confusing, and the plots are almost impossible to read. For the experiments, I do not understand the purpose of the baseline, as I understand that they are not performing a preemptive algorithm for computing property paths (also, I get this from the HTTP calls plot). The idea is to understand how fast/slow is your approach with comparison to standard techniques? Please clarify this!

I think that your approach can be so useful in practice and is indeed interesting, but the presentation of your work can improve a lot. For now I recommend to weak reject this paper, but the decision may change in base to the answer of the authors.

**Anonymity:**

Yes, I would like my review to remain anonymous.

**Reuse And Availability:**

2: Low

**Strong Points:**

- The problem is indeed interesting and the solution has potential.
- Their experimental results look ok.

**Subreviewer:**

I submitted this review.

**Weak Points:**

- The paper is hard to read.
- Some figures can be improved in order to be a help for the reader in understanding the main ideas.
- The lack of a complete example that explain the approach, together with an easier graph to follow a step by step presentation.

---

> ### Author Rebuttal · Authors · 2021-01-29
>
> We thank the reviewer for their positive comments.
>
> Although the problem is well-motivated, the introduction is not fluid and the related work has room for improvement. For instance, there is no discussion of some proposals such as those discussed in "Evaluating Navigational RDF Queries over the Web" (Baier, Daroch, Reutter & Vrgoc, 2017).
>    * Baier2017 proposes a nice alternative to DFS/BFS graph traversal based on A*. However, we believe that  A* is not preemptable (due to its space complexity). We can add Baier2017 to graph traversal approaches.
>
> Also, the authors claim that some strategies used by SPARQL endpoints rely on Recursive Queries; it would be great to have a detailed list of the strategies used by the most famous engines (Jena, Virtuoso, Blazegraph). As far as I know, none of them uses Recursive Queries.
>    * We apologize for the confusion; recursive queries or graph traversal algorithms are common approaches for computing transitive closures, we removed “implemented in SPARQL endpoints”.
>
>
> Also, at some point, the authors claim that nested regular expressions (such as (ab*)*) are rare and not of interest. I disagree with this opinion, since use cases such as those studied in "Relative Expressiveness of Nested Regular Expressions" (Barceló, Pérez & Reutter, 2012) and some other papers show that these expressions are in fact important.
>    * We do not completely agree with the reviewer. We wrote “In this paper, we focus on property path expressions without nested transitive closures, for example, a property path like (ab*)* is not considered and are very rare in practice [5, 6]”. This comes from the results of analyzing different sparql endpoints query logs [5, 6]. Definitely, expressions with nested stars are interesting and challenging future work.
>
> In section 4, the authors present their algorithms to solve property path queries with web preemption. However, I think that this section is hard to read and can be improved. For instance, there are some examples, but it would be great to have an example that connects all the parts of the strategy to understand the complete workflow of the approach.
>    * Figure 4 connects all the parts of the approach. It evaluates query Q3 on D. During the evaluation, serverEval(Q3) executes the query plan described in Figure 3 where pPTC operator is called. We improved the description of Figure 4 to present the complete workflow of the approach.
>
> Also, there are practical issues that are not discussed. When a user asks for a property path query to an endpoint, what happens when the quanta is over? The user receives some partial results or he has to wait for the endpoint to have all the results?
>    * When a quantum is over, all the results obtained during the quantum are sent to the user. Then, the server resumes immediately the next waiting query on the server.
>
> Also, I would like to know how long are the times between distinct quanta in practice.
>    * Time to save the state of a query under execution and to resume the next waiting query. In practice, this represents a few ms (less than 10ms). This is well described in [11].
>
> With respect to the queues of remaining nodes. They are stored at the server or at the client? How is the interaction between client and server? This can be explained better.
> * The server is purely stateless. All control tuples and results are stored client-side. We clarified this.
>
> Also, please clarify Figure 4, it is not useful to understand your approach. Moreover, it would be great to discuss what happens in case of concurrent queries done by distinct users.
>    * Figure 4 represents the execution of one iteration of algorithm 3 for Q3. All ‘eval’ calls of Figure 4 correspond to ‘serverEval’ calls in algorithm 3. We reworked the explanations of Figure 4 to present the complete workflow of the approach.
>
> Then, regarding the experimental section, I wanted to take a look at the implementation, but the link was broken. it would be great to have access to the queries generated by the benchmark, as sometimes the queries created by GMark does not make sense.
> * We regret this inconvenience. The repository is up now at https://github.com/JulienDavat/property-paths-experiments
>
> For the experiments, I do not understand the purpose of the baseline, as I understand that they are not performing a preemptive algorithm for computing property paths (also, I get this from the HTTP calls plot). The idea is to understand how fast/slow is your approach with comparison to standard techniques? Please clarify this!
> * Compared to Virtuoso/Jena without quotas, our proposal has clearly an overhead. The objective was to quantify this overhead. Surprisingly, we observed that we can compete with Jena and, for some queries, with Virtuoso. This demonstrates that it can be possible to achieve high performances while ensuring complete query results.

---

> > ### Comment · AnonReviewer4 · 2021-02-03
> > **Answer**
> >
> > Thank you for your answer. As I said in my updated review, I think that your paper presents a solid piece of work and it is good enough to be published. However, the presentation can be improved and not only by updating the Figure 4. I encourage you to improve the writing of your work having in mind all the comments raised by the reviewers.

---

### Official Review · AnonReviewer2 · 2021-01-15
**Extension for property paths to previous work**

**Rating:** 1
**Confidence:** 4
**Impact:** 2
**Design And Technical Quality:** 4

**Review:**

The paper presents an approach to evaluate SPARQL property path queries in the presence of quotas of SPARQL endpoints that prohibit complete results. Their approach follows their preemption approach to suspend query evaluation.

The paper is of high quality and clarity.

Regarding originality, it is a continuation of previous work for another feature of SPARQL, so not groundbreakingly new.

Regarding significance, I don't know about the size of the "customer base" who needs to be able to work around other peoples' quotas instead of setting up the triple store themselves.

## after rebuttal:
I have read the other reviewers' comments and the rebuttals. Thanks! I stick with my borderline positive verdict. On the plus side, the technical thoroughness as acknowledged also by the other reviews; on the negative side see my remarks on originality and significance.

**Anonymity:**

Yes, I would like my review to remain anonymous.

**Reuse And Availability:**

4: High

**Strong Points:**

* The paper is well-structured and well-written
* The approach seems to be thoroughly thought through

**Subreviewer:**

I submitted this review.

**Weak Points:**

* There is a lot of work that deals with limitations and quotas of SPARQL endpoints. In an evaluation I would be interested to see whether it is *overall* good to impose such quotas in the first place. Now that we have these kinds of clients that can work around the quotas, did the SPARQL endpoint save resources at the end? What is the *overall* performance? E.g. how many resources does the system spend, ie. client and server spend *combined*?

Minor:
* What makes a client a _smart_ client?

---

> ### Author Rebuttal · Authors · 2021-01-29
>
> We thank the reviewer for their positive evaluation.
>
> Regarding originality, it is a continuation of the previous work for another feature of SPARQL, so not groundbreakingly new.
>    * In this paper, we demonstrated that it is possible to execute path queries with a limited depth of exploration and restart the execution from where we have cut the exploration. This original technique allows to execute property path queries online and get complete results with low data transfer and execution time comparable to Jena.
>
> Regarding significance, I don't know about the size of the "customer base" who need to be able to work around other peoples' quotas instead of setting up the triple store themselves.
>    * We agree that reinstalling data from dumps is a workaround for quotas of public SPARQL endpoints. However, the size of the last wikidata dump is now up to 117Go (compressed), the ingestion time of such a dataset is estimated around a few days and local data have freshness issues. Smart client approaches allow us to query data online and get complete results.
>
>
> There is a lot of work that deals with limitations and quotas of SPARQL endpoints. In an evaluation, I would be interested to see whether it is overall good to impose such quotas in the first place.
>    * Without quotas, the endpoint is unfair, i.e., few people executing long-running queries will block the server. According to DBpedia “A Fair Use Policy is in place in order to provide a stable and responsive endpoint for the community.”
>
> Now that we have these kinds of clients that can work around the quotas, did the SPARQL endpoint save resources at the end?
>    * Compared to endpoints, a smart-client execution of the same workload may consume more resources (as shown in the experiments), but queries are guaranteed to terminate. If an endpoint kills your query after the 60s and you really want complete results, you are forced to download and reinstall all data locally and run the query yourself. In this way, a smart-client saves resources.
>
> What is the overall performance? E.g. how many resources does the system spend, ie. client and server spend combined?
>    * When we measure execution time in the paper, we combine the time spent on the client and the time spent on the server, i.e. the overall performance. The comparison with Jena/Virtuoso allows us to evaluate the overhead of the approach.
>
> What makes a client a smart client?
>    * Because a part of the query processing is performed on the client. In this paper, we succeed to compute property path queries without any joins on the smart client, the smart client restarts queries from frontiers nodes until completion.

---

### Official Review · AnonReviewer5 · 2021-01-17
**Processing SPARQL Property Path Queries Online with Web Preemption**

**Rating:** 1
**Confidence:** 4
**Impact:** 4
**Design And Technical Quality:** 3

**Review:**

This paper proposes a method on how to execute SAPRQL property path queries using SAGE-based client serve scenario. The extension is based on a  partial transitive closure operator (PTC) on a depth limited search algorithm. The proposed approach is compared with the author's previous approach [1], virtuoso, Jena-Fuseki by using synthetic data and queries generated using gMark framework. Experiments show promising results as compared to the selected approaches in terms of query runtime performance.

Overall, I really liked the paper. The different configurations are well justified. However, I have some  concerns.

1. This is an extension of the author's previous work on Web Preemption, i.e. addition into the SAGE server. The addition is the implementation of the PTC operator to execute SPARQL property path queries. As such, I am slightly curious about the *sufficient scientific contribution/novelty* of this work.

2. gMark, a synthetic data and query generator was used to select the datasets and queries used in the evaluation. Since the SPARQL query logs of many SPARQL endpoints are already available, the author could have used real-world datasets and the property path queries (selected from query logs) to test the proposed approach in real-world settings. For example, the DBpedia and Wikidata query logs contain sufficient property path queries which could have been used in the evaluation. The FEASIBLE benchmark generation framework could be used to select well representative property path queries in the benchmark.

3. Unfortunately, the provided github repo is not available.

**After Rebuttals**

Thank you for the clarifications.  I am still curious about the scientific novelty of the paper.  Nevertheless, the work has great potential and I encourage authors to continue their good work in this direction.


**Anonymity:**

No, I would like my review to be deanonymized.

**Reuse And Availability:**

1: Very low

**Strong Points:**

1. The topic is relevant to the conference.

2. The evaluation result is promising.



**Subreviewer:**

I submitted this review.

**Weak Points:**

1. The scientific novelty of the paper is a bit limited.

2.  The benchmark could have been selected from real-world datasets and queries.

3. Github repo is unavailable.

---

> ### Author Rebuttal · Authors · 2021-01-29
>
> We thank the reviewer for their positive evaluation.
>
> The scientific novelty of the paper is a bit limited.
>    * In this paper, we demonstrated that it is possible to execute property path queries with an upper bound on the exploration depth and restart the execution from where the exploration has been cut. Unlike the state of art, we show that processing property path queries while generating complete results can be done without doing any joins on the client-side. As all joins are performed on the server-side, we outperform existing smart client approaches and deliver performances comparable with Jena.
>
> The benchmark could have been selected from real-world datasets and queries.
>    * We have considered many real-world queries extracted from the wikidata logs but most property path queries only consider simple path patterns and hierarchies (i.e., subclassof*). As pointed out in [2], clique and dense graphs are challenging for property path queries and especially for our PTC operator. We finally selected the gmark benchmark for its ability to generate hierarchies, cycles, and dense graphs with a corresponding and challenging workload of queries. With this setup, the results of this paper can also be compared with the results of [1].
>
> Github repo is unavailable.
>    * We regret this inconvenience. The repository is up now at https://github.com/JulienDavat/property-paths-experiments

---

### Decision · Program_Chairs · 2021-02-23

**Decision:**

Accept

**Comment:**

The majority of the reviewers recommended that the paper is included in the proceedings. However, there is a general consensus that the value of this work would increase significantly if the authors address many problems/ambiguities in the presentation.
Such issues range from typos to more structural problems and they have a big impact on the readability of the paper. This is a pity because the reviewers recognise that the research is interesting and valuable for the community.

Fortunately, some reviewers report a detailed list of problems. Our final recommendation about the acceptance is inline with the one of the reviewers, but we strongly encourage the authors to follow the suggestions provided by the reviewers and improve the camera-ready accordingly.